# Maximum-Entropy Priors with Derived Parameters in a Specified Distribution

**DOI:** 10.3390/e21030272

**Published:** 2019-03-12

**Authors:** Will Handley, Marius Millea

**Affiliations:** 1Astrophysics Group, Cavendish Laboratory, J.J.Thomson Avenue, Cambridge CB3 0HE, UK; 2Kavli Institute for Cosmology, Madingley Road, Cambridge CB3 0HA, UK; 3Gonville & Caius College, Trinity Street, Cambridge CB2 1TA, UK; 4Institut dAstrophysique de Paris (IAP), UMR 7095, CNRS UPMC Universit Paris 6, Sorbonne Universits, 98bis Boulevard Arago, F-75014 Paris, France; 5Institut Lagrange de Paris (ILP), Sorbonne Universits, 98bis Boulevard Arago, F-75014 Paris, France

**Keywords:** maximum entropy, Bayesian inference, prior, derived distribution, neutrino hierarchy

## Abstract

We propose a method for transforming probability distributions so that parameters of interest are forced into a specified distribution. We prove that this approach is the maximum-entropy choice, and provide a motivating example, applicable to neutrino-hierarchy inference.

## 1. Introduction

In Bayesian analysis, a simple prior on inference parameters can induce a nontrivial prior on critical physical parameters of interest. This arises, for example, when estimating the masses of neutrinos from cosmological observations. Here, three parameters are inferred corresponding to the mass of each of the three neutrino species, $\left(m_{1},m_{2},m_{3}\right)$. Cosmological observations, however, are mainly sensitive to their sum, $m_{1}+m_{2}+m_{3}$. Simple priors, for example, log-uniform priors on individual masses, can induce undesired informative priors on their sum [1].

Another example arises in nonparametric reconstructions. Here, one infers underlying physical function from the data, where the data are a reprocessing of the target function by some physical or instrumental transfer function. Typical approaches involve decomposing the target function into bins, principal component eigenmodes, or generally into any other basis functions. Simple priors on the amplitudes of basis functions can lead to undersized priors on physical quantities derived from the target function. Consideration of these effects is particularly important, for example, when reconstructing the history of cosmic reionization [2].

A natural remedy is to importance-weight the original prior such that the nontrivial distribution on the parameter of interest is transformed to a more desirable one. In this paper, we show that this natural approach is the maximum-entropy prior distribution [3]. Often, the more desirable prior is a uniform distribution, but our proof also holds for any desired target distribution. Our observation provides a powerful justification for the natural solution, as it is the distribution that assumes the least information, and is therefore particularly appropriate for choosing priors [4].

In Section 2, we demonstrate the key ideas with a toy example before providing a rigorous proof in Section 3. We then apply these ideas to a more complicated example, appropriate for constructing priors on neutrino masses, in Section 4.

## 2. Motivating Example

We begin with a simplified example. Consider a system with two parameters (a,b), with a uniform distribution q(a,b) on the unit square. Analogous to the sum of neutrino masses mentioned earlier, suppose that a derived parameter, c=a+b, is of physical interest. Effective distribution q(a+b) is not uniform, but instead symmetric and triangular between 0<c<2, as graphically illustrated in the left-hand side of Figure 1. If one wished to construct a distribution p(a,b) that was uniform in a+b, one could do so by dividing out the triangular distribution:(1)$$p\left(a,b\right)=\frac{q(a,b)}{q(a+b)}.$$

The resulting transformed distribution is illustrated in the right-hand side of Figure 1. More weight is given to low and higher values of *a* and *b*, so that the tails of triangular distribution q(a+b) are counterbalanced. This comes at the price of altering the marginal distributions of *a* and *b*, which become p(a)=−log[a(1−a)]/2 (similarly for *b*), but which now give a uniform prior, p(a+b). The transformation can be viewed as an importance weighting of the original distribution, and is intuitively the simplest way to force p(a+b) to be uniform.

The aim of this paper is to show that the above intuition is well-founded, as (Equation 1) is in fact the maximum-entropy solution. The entropy of a distribution p(x) with respect to an underlying measure q(x) is:(2)$$H\left(p|q\right)=\int d\;\;x\;p\left(x\right)\log\left[\frac{q(x)}{p(x)}\right].$$

The maximum-entropy approach [6,7] finds distribution *p* that maximises *H*, subject to user-specified constraints. As it maximises entropy, solution *p* is generally interpreted as the distribution that assumes the least information given the constraints.

In the next section, we show that (Equation 1) is the maximum-entropy solution, subject to the constraint that p(a+b) is uniform. We further generalize to a derived parameter that can be any arbitrary function of the original parameters, for which the desired distribution is in general nonuniform.

In a more usual maximum-entropy setting, user-applied constraints typically take the form of either a domain restriction such as x∈[−1,1] or x>0, or linear functions of distribution *p*, such as a specified mean μ=∫xp(x)d x, or variance $\sigma^{2}=\int {\left(x-\mu \right)}^{2}p\left(x\right)d\;\;x$. In this work, our constraints contrast with the traditional approach in that, instead of a discrete set of constraints, by demanding that a derived parameter has a distribution in a specified functional form, our constraints form a continuum. In other words, instead of a discrete set of Lagrange multipliers, one must introduce a continuous Lagrange multiplier function.

## 3. Mathematical Proof

**Theorem** **1.**
*If one has a D-dimensional distribution on parameters x with probability density function q(x) along with a derived parameter f defined by a function f=f(x), then the maximum-entropy distribution p(x) relative to q(x) satisfying the constraint that f is distributed with probability density function to r(f) is:*
(3)$$p\left(x\right)=\frac{q(x)r(f(x))}{P(f(x)|q)},$$
*where P(f|q) is the probability density for the distribution induced by q on f=f(x).*


**Proof.** If we have some function f(x) defining a derived parameter f=f(x), then cumulative density function C(f|p) of f=f(x) induced by *p* can be expressed as a *D*-dimensional integral over the region Ω(f)={x:f(x)<f} with *D*-dimensional volume element d x:
(4)$$C\left(f|p\right)=\int_{f(x)<f}d\;\;x\;p\left(x\right).$$Differentiating (Equation 4) with respect to *f* yields the probability density function of *f* induced by *p*, which via the Leibniz integral rule can be expressed as a (D−1)-dimensional integral over the boundary surface ∂Ω(f)={x:f(x)=f}, with the induced (D−1)-dimensional volume element d S(x):
(5)$$P\left(f|p\right)=\frac{d\;\;}{d\;\;f}C\left(f|p\right)\equiv \int_{f(x)=f}d\;\;S(x)\;p\left(x\right).$$We aim to find distribution *p* that maximises entropy H(p|q) from (Equation 2), subject to the constraint that P(f|p) takes a given form with probability density r(f) and cumulative density c(f):
(6)$$C\left(f|p\right)=c\left(f\right)\;\Leftrightarrow \;P\left(f|p\right)=r\left(f\right)=c^{\prime }\left(f\right)$$The solution can be obtained via the method of Lagrange multipliers, wherein we maximise the functional *F*:
(7)$$\begin{array}{cc} F(p) & =H\left(p|q\right)-\lambda \int p\left(x\right)d\;\;x-\int d\;\;f\;\mu \left(f\right)\int_{f(x)<f}d\;\;x\;p\left(x\right), \end{array}$$
subject to normalisation and distribution constraints
(8)$$\begin{array}{c} \int p\left(x\right)d\;\;x=1, \end{array}$$
(9)$$\begin{array}{c} \int_{f(x)<f}p\left(x\right)d\;\;x=c\left(f\right)\;\Leftrightarrow \;\int_{f(x)=f}d\;\;S(x)\;p\left(x\right)=r\left(f\right). \end{array}$$Here, we introduced a Lagrange multiplier λ for the normalisation constraint (Equation 8), and a continuous set of Lagrange multipliers μ(f) for the distribution constraints (9).Functionally differentiating (Equation 7) yields:
(10)$$\begin{array}{cc} 0 & =\frac{\delta F}{\delta p(x)}=-1+\log\frac{p(x)}{q(x)}-\lambda -\int_{f(x)<f}d\;\;f\;\mu \left(f\right), \end{array}$$
(11)$$\begin{array}{cc} \Rightarrow p(x) & =q\left(x\right)e^{1+\lambda +\int_{f(x)<f}d\;\;f\;\mu \left(f\right)}=q\left(x\right)M\left(f\left(x\right)\right), \end{array}$$
where in (Equation 10) we have used the fact that:
(12)$$\frac{\delta }{\delta p(x)}\int_{f(x^{\prime })<f}d\;\;x^{\prime }\;p\left(x^{\prime }\right)=\left\{\begin{array}{cc} 1 & :f(x)<f\\ 0 & :\mathrm{otherwise}, \end{array}\right.$$
and, in (Equation 11), defined the new function:
(13)$$\log M\left(g\right)=1+\lambda +\int_{g<f}d\;\;f\;\mu \left(f\right).$$All that remains to be done is to determine *M* from Constraints (Equation 8) and (9). Taking the right-hand form of distribution Constraint (9), and substituting in p(x)=q(x)M(f(x)) from (Equation 11), we find:
(14)$$\begin{array}{cc} r(f) & =\int_{f(x)=f}d\;\;S(x)\;q\left(x\right)M\left(f\left(x\right)\right)=M\left(f\right)\int_{f(x)=f}d\;\;S(x)\;q\left(x\right)=M\left(f\right)P\left(f|q\right), \end{array}$$
where we have used the fact that M(f(x)) is constant over the surface f(x)=f, and Definition (Equation 5) for a constrained probability distribution function. We now have the form of *M* to substitute into (Equation 11), yielding Solution (Equation 3). □

Result (Equation 3) is precisely what one would expect. The distribution that converts q(x) to one, which instead has f=f(x) distributed according to r(f), is found by first dividing out the distribution on *f* induced by *q*, and then modulating by desired distribution r(f).

Provided that r(f) is correctly normalised, Expression (Equation 3) automatically satisfies normalisation Constraint (Equation 8):(15)$$\begin{array}{cc} \int d\;\;x\;\frac{q(x)r(f(x))}{P(f(x)|q)} & =\int d\;\;f\int_{f(x)=f}d\;\;S(x)\frac{q(x)r(f(x))}{P(f(x)|q)}\\ & =\int d\;\;f\frac{r(f)}{P(f|q)}\int_{f(x)=f}d\;\;S(x)q\left(x\right)=\int d\;\;f\;r\left(f\right)=1. \end{array}$$

In the above, we first split the volume integral into a set of nested surface integrals, drew out the functions that were constant over the surfaces, applied the definition of induced probability density P(f|q), and then used the normalisation of *r*. A similar manipulation may be used to confirm that functional Form (Equation 3) satisfies distribution Constraint (9).

The proof may be generalised to multiple derived parameters without modification, simply taking f=f(x) to represent a vector relationship, and the cumulative distribution functions to be their multiparameter equivalents.

## 4. Example: Neutrino Masses

In the past year, there has been interest in the cosmological and particle-physics community regarding the correct prior to put on neutrino masses. Simpson et al. [8] controversially claimed that, with current cosmological parameter constraints ($\sum_{\nu }m_{\nu }<0.13\;\mathrm{eV}$ [9,10]), the normal hierarchy of masses was strongly preferred over an inverted hierarchy, in contrast with the results of Vagnozzi et al. [11]. Later, Schwetz et al. [1] showed that the controversial claim was mostly due to a nontrivial prior that had been put on the neutrino masses. Since then, other choices of prior have been proposed by Caldwell et al. [12], Long et al. [13], Gariazzo et al. [14] and Heavens and Sellentin [15], which reduce the strength of the claim.

Using our methodology, a possible alternative prior to put on the masses can be constructed. Typically, one chooses a broad independent logarithmic prior on each of the masses of the three neutrinos $\left(m_{1},m_{2},m_{3}\right)$. However, cosmological probes of the neutrino masses typically place a constraint on the sum of the masses $m_{1}+m_{2}+m_{3}$. Simple logarithmic priors on the masses place a nontrivial prior on their sum. Using our approach, we can transform the initial distribution into one that has more reasonable distribution on the sum of the masses. Such considerations can be particularly important when determining the strength of cosmological probes.

A concrete example is illustrated in Figure 2. As the original distribution, we take an independent Gaussian prior on the logarithm of the masses. This induces nontrivial distribution on the sum of the masses, approximately log-normal, but with a shifted centre. If one demands that the sum of the masses is instead centred on zero, then the maximum-entropy approach creates a distribution with tails toward low masses in order to compensate for the upward shift in the distribution of the sum of the masses. This tail enters a region of parameter space that would be completely excluded by the original prior; thus, choosing the transformed prior could influence the strength of a given inference on the nature of the neutrino hierarchy. It should be noted that we are not advocating this as the most suitable prior to put on neutrino masses, but merely to show that you may use our procedure to straightforwardly transform a distribution, should one wish to put a flat prior on the sum of the masses. A more physical cosmological example in the context of reionization reconstruction can be found in Millea and Bouchet [2].

## 5. Conclusions

In this paper, we proposed an approach for transforming probability distribution to force a derived parameter into a specified distribution. One importance-weights the original distribution by dividing out the induced distribution on the parameter of interest, and reweights by the desired distribution. We proved that the resulting distribution is the maximum-entropy choice. Finally, we provided some motivating examples.

## Figures and Tables

**Figure 1 entropy-21-00272-f001:**
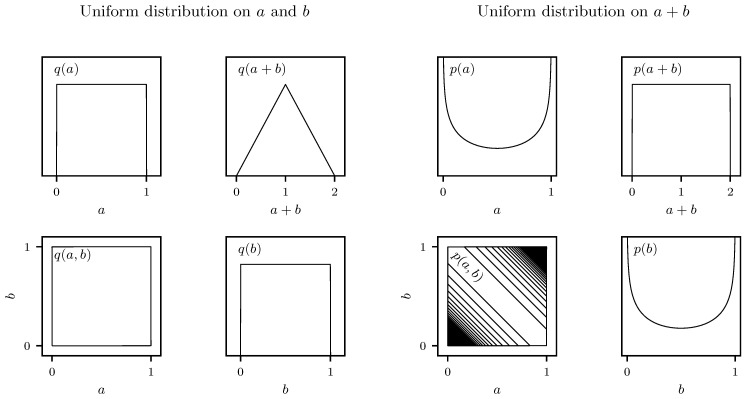
(Left-hand panels) uniform distribution on two parameters q(a,b) inducing triangular distribution on their sum q(a+b). (Right-hand panels) constructing new distribution by dividing out triangular distribution p(a,b)=q(a,b)/q(a+b) renders a uniform distribution induced on the sum p(a+b). Figures were constructed from the analytic forms of the distributions in Python using the Matplotlib package [5].

**Figure 2 entropy-21-00272-f002:**
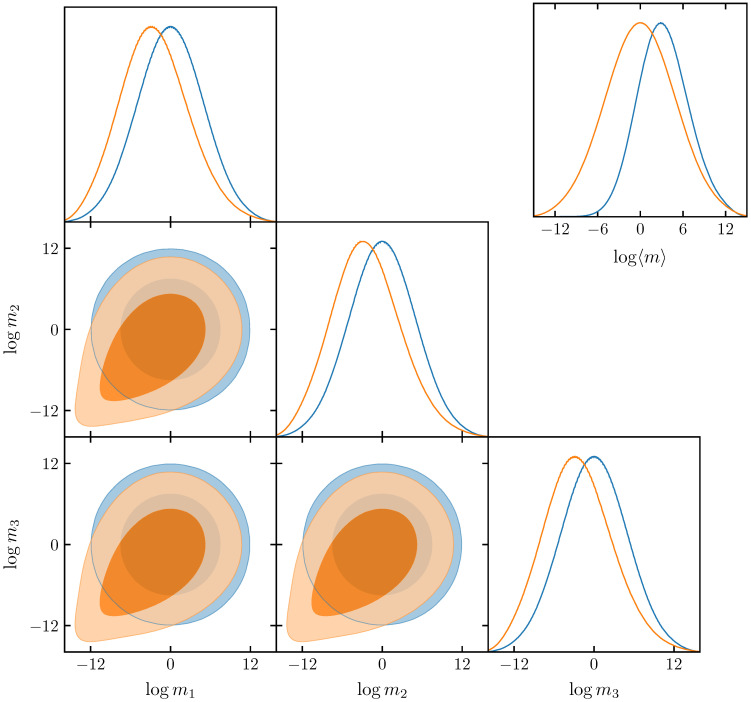
Distribution *q* (illustrated in blue) is defined as a three-dimensional spherical Gaussian on the logarithm of parameters $m_{1}$, $m_{2}$, $m_{3}$, centred on zero with a standard deviation of five log units on each parameter. Nontrivial distribution is induced on the mean of masses $\langle m\rangle =\frac{1}{3}\left(m_{1}+m_{2}+m_{3}\right)$, which is approximately log-normal, but with a shifted centre and width. If one demands that the mean of the masses is log-normal centred on zero with width five, as for the original individual masses, then the maximum-entropy approach creates the distribution *p*, illustrated in orange. Parameters are forced to have a tail toward low values in order to compensate for the upward shift in *q*-mean distribution.

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
