# Peer review of "Maximum-Entropy Priors with Derived Parameters in a Specified Distribution"

_entropy, 2019, doi:10.3390/e21030272_

Round 1

Reviewer 1 Report

The paper is well written and deserve to be published in MPDI Journal.

By the way I will prefer to have a more precise description of section 2. Motivating example; i.e. how to obtain the figures reported in figure 1.

A second improvement could be to quantify the value of the neutrino masses according to the last experimental results. 

Author Response

We thank the reviewer for their positive review,

> By the way I will prefer to have a more precise description of section 2. Motivating example; i.e. how to obtain the figures reported in figure 1.

We have included an additional description in the caption of figure 1 detailing how it was produced.

> A second improvement could be to quantify the value of the neutrino masses according to the last experimental results. 

We have included a quote of the current constraints on neutrino masses, with the corresponding references.

Best,

Will & Marius

Reviewer 2 Report

I would recommend the paper for publication after clarifying the following point:

What is the relation between the author's method and the minimal of cross-entropy principle (e.g. Jaynes, Phys.Rev. 108 (1957) 171). The optimized function is the same, but whereas the constraint in the authors method is a strong one amounting to the probability density function itself $r$, the constraint in Jaynes approach is 'weaker' constraining just moments (expectation values). Is one method a special case of the other? or the two methods are unrelated in view that the subsets overwhich to optimize are different?   

Author Response

We thank the reviewer for their positive review.

With regards to their query, we have added a section of  the text, along with the citation they refer to:

  In a more usual maximum entropy setting, the user-applied constraints typically take the form of either a domain restriction such as $x\in [-1,1]$ or $x>0$, or linear functions of the distribution $p$, such as a specified mean $\mu = \int x p(x) \d{x}$, or variance $\sigma^2 = \int (x-\mu)^2 p(x) \d{x}$. In this work our constraints contrast with the traditional approach in that instead of a discrete set of constraints, by demanding that a derived parameter has a distribution in a specified functional form, our constraints form a continuum. In other words, instead of a discrete set of Lagrange multipliers, one must introduce a continuous Lagrange multiplier function.

Best,

Will & Marius

Reviewer 3 Report

Dear Editor,

The Authors present a novel method suitable to importance weight the prior distribution of a given set of parameters, m_i with i=1 ... N, in such a way that a given function of the latter, f(m_i i=1,N), has the desired behavior. The manuscript rigorously proves that the prior distribution for m_i given f(m_i) is the one that maximizes the entropy. 

I find the paper scientifically sound and well written. However, before I recommend it for publication in Entropy, I would like the Authors to clarify a couple of points of their work:

i) As an outsider of the field, in the specific example on the masses of the neutrinos, it is not clear to me why a flat distribution of their sum is obviously the desired one. Let us suppose that one wants to to put a prior on the possible sums of two dice. If the two dice are fair and independent, the prior distribution on the outcome for a single dice is flat between 1 and 7. This yields a non-flat prior on the probability for the sum of two dices that has a peak in 7. Assuming a flat distribution on the sum, would in this case enforce a non-physical distribution for the single-dice outcome. I would appreciate if the Authors could better clarify the physical reasons allowing for a non log-uniform priors on the single neutrino masses. 

ii) Although intuitively sound, the Author should clarify how to go from Eq. (4) to Eq. (5). In particular, more details should be provided on the transformed volume element dS(x). 

Author Response

We thank the reviewer for their positive review.

i) As an outsider of the field, in the specific example on the masses of the neutrinos, it is not clear to me why a flat distribution of their sum is obviously the desired one. Let us suppose that one wants to to put a prior on the possible sums of two dice. If the two dice are fair and independent, the prior distribution on the outcome for a single dice is flat between 1 and 7. This yields a non-flat prior on the probability for the sum of two dices that has a peak in 7. Assuming a flat distribution on the sum, would in this case enforce a non-physical distribution for the single-dice outcome. I would appreciate if the Authors could better clarify the physical reasons allowing for a non log-uniform priors on the single neutrino masses. 

As is referenced in the paper, there has been a lot of discussion in the literature on the issues associated with 'box' log-uniform priors. The difference between the neutrino case and the dice rolling case that the reviewer refers to is that there is no easily defined lower or upper bound in log space for the neutrino masses. The edges of the box in log space induce effects that do not correspond to the intuition of 'no knowledge of scale', and the purpose of the log-gaussian prior is to smooth these effects.

It should be noted that we are not necessarily advocating this as the correct prior to put on neutrino masses, merely that if you wanted to enforce a specific distribution on the sum of the masses, then our procedure makes this easy to do, and provides a theoretical underpinning to justify the reweighting of the prior. We have added a sentence to the final paragraph of section 4 to this effect.

If necessary, if the reviewer requires it we are happy to expand in further detail in the paper, but we originally omitted such expansion as it is well-covered in the literature, and were aiming for this paper to be of letter length.

ii) Although intuitively sound, the Author should clarify how to go from Eq. (4) to Eq. (5). In particular, more details should be provided on the transformed volume element dS(x). 

We have now clarified this step in the text as an application of the Leibniz integral rule, and added text to make clear what the notation for the volume element dS(x) represents.

Best,

Will & Marius